# Brain Delivery of Single-Domain Antibodies: A Focus on VHH and VNAR

**DOI:** 10.3390/pharmaceutics12100937

**Published:** 2020-09-30

**Authors:** Elodie Pothin, Dominique Lesuisse, Pierre Lafaye

**Affiliations:** 1Antibody Engineering Platform, Structural Biology and Chemistry Department, Institut Pasteur, 75015 Paris, France; elodie.pothin@pasteur.fr; 2Tissue Barriers, Rare and Neurological Diseases TA Department, Sanofi, 91161 Chilly-Mazarin, France

**Keywords:** variable domain of heavy-chain antibody (VHH), variable new antigen receptor (VNAR), blood–brain barrier (BBB), drug delivery, single-domain antibody

## Abstract

Passive immunotherapy, i.e., treatment with therapeutic antibodies, has been increasingly used over the last decade in several diseases such as cancers or inflammation. However, these proteins have some limitations that single-domain antibodies could potentially solve. One of the main issues of conventional antibodies is their limited brain penetration because of the blood–brain barrier (BBB). In this review, we aim at exploring the different options single-domain antibodies (sDAbs) such as variable domain of heavy-chain antibodies (VHHs) and variable new antigen receptors (VNARs) have already taken to reach the brain allowing them to be used as therapeutic, diagnosis or transporter tools.

## 1. Introduction

Monoclonal antibodies have been of common use for several years now for the treatment of several diseases such as cancers or inflammation. Their high affinity and selectivity for their targets, along with their potential to reach intractable or difficult targets such as protein–protein interactions or aggregated proteins, make them tools of choice in several indications. However, these proteins also have limitations. As imaging tool, their long half-lives (around several days or weeks) make them inappropriate because of their low clearance from the organism [1]. Diffusion of conventional antibodies is restricted in tissues due to their large size (150 kDa) and in particular by the blood–tumor barrier (BTB) limiting access to the tumor center [2]. More specifically, their use for brain diseases (glioblastoma, neurodegeneration, etc.) has been hampered by the blood–brain barrier (BBB) that limits their access to the brain [3]. The brain is a highly protected tissue, with extremely tightly sealed endothelial cells equipped with many efflux transporters and metabolic systems, preventing molecules’ penetration, even more so large hydrophilic compounds such as antibodies [4,5]. Some of the limits of antibodies could potentially be overcome by single-domain antibodies (sDAbs) such as variable domain of heavy-chain antibodies (VHHs), also called Nanobodies^®^, or variable new antigen receptors (VNARs) due to their much lower size and different pharmacokinetic properties. Several endocytic mechanisms such as receptor-mediated transcytosis, adsorptive transcytosis or macropinocytosis have been reported for immunoglobulin Gs (IgGs) [6], and several strategies have been used to increase brain exposure of biotherapeutics [7]. A few recent reports have reviewed single-domain antibodies directed against brain targets [8] and optimization of nanobodies to treat neurodegenerative disorders [9]. The object of the present review is to summarize the state of the art regarding brain exposure of single-domain antibodies with a focus on VHHs and VNARs. We show that some VHHs can cross the blood–brain barrier (BBB) directly or be delivered indirectly and act either on their own or by delivering an active payload into the brain. We first describe VHHs and VNARs with their characteristics making them unique proteins. Then, we discuss the different mechanisms VHHs and VNARs can use to reach the brain. Finally, we describe the use of brain penetrating sDAbs as therapeutic, diagnosis or transporter tools.

## 2. Single-Domain Antibodies and Their Properties: Variable Domain of Heavy-Chain Antibodies (VHHs) and Variable New Antigen Receptors (VNARs)

In addition to conventional antibodies, made of two heavy and two light chains, camelids and sharks produce unusual antibodies composed only of heavy chains. Their variable antigen-binding domain is formed by a single-domain. In *Camelidae*, it is designated by VHHs for variable domain of heavy-chain antibodies and in some cartilaginous fishes such as sharks [10], it is designated by VNARs for variable new antigen receptor.

### 2.1. VHH

VHHs are the variable domains of heavy-chain antibodies (HCAb) found in *Camelidae*. This family is composed of Vicugna, Alpaca, Llama, Camel and Dromedary. As other mammals, they express conventional antibodies composed of two heavy chains and two light chains. In addition, they also express non-conventional antibodies named heavy-chain antibodies devoid of CH1 domain and light chain (Figure 1). HCAb are present for IgG2 and IgG3 isotypes. These antibodies are composed of two heavy chains divided in three domains each: CH3–CH2–VHH. The molecular weight of these HCAb is 90 kDa, smaller than the 150 kDa of conventional antibodies [11].

The variable domain of HCAb corresponding to the paratope recognizing the antigen is called VHH. It is also named *Nanobody*^®^, a name registered by Ablynx, a Sanofi company working specifically on these proteins (NANOBODY Trademark of ABLYNX N.V.–Registration Number 5098047–Serial Number 85573029: Justia Trademarks Available online: http://trademarks.justia.com/855/73/nanobody-85573029.html (accessed on 11 June 2020)). The mean molecular weight is 15 kDa, tenfold smaller than a conventional antibody. This variable domain can be expressed on its own and still recognize the antigen [12]. These proteins have specific properties making them unique tools. They have a high identity rate with mouse and human variable domain of conventional antibodies heavy chain (VH) around 80% [13], and, even if they come from a different species than the one treated, they are weakly immunogenic in mice [14,15]. With this criterion, VHHs can be used in humans. In 2018, the first VHH named caplacizumab was approved in Europe and then in the US. This VHH was developed against von Willebrand factor and is available for patients with acquired thrombotic thrombocytopenic purpura [16,17]. VHHs low immunogenicity has been evaluated with native VHHs. In this review, we also deal with mutated or complexified VHHs that might have different immune responses that will have to be checked before clinical use. Compared to conventional antibodies, VHHs are more stable and can refold after denaturation [18]. They are also highly soluble owing to hydrophilic amino acids mutated compared to VH [19]. Because of their small size, VHHs have a short half-life between 0.5 and 2 h [14,15] compared to several weeks for conventional antibodies [20]. To be used as therapeutics, they can be linked to a life extension module [21,22]. The affinity is similar to antibodies, most of the time in the nanomolar range [23]. In addition, VHHs can reach other epitopes than conventional antibodies. For example, VHH can recognize cryptic epitopes [24] normally hidden from the immune system and not recognized by immune cells in a normal process. VHH are easily produced in prokaryotic or eukaryotic expression systems [25] and can be obtained in decent yield in *Pichia Pastoris* [26] or *Escherichia coli* [27].

Their small size allows them to diffuse more than four times better in tissues [28,29] and tumors [14] than antibodies. However, half-life-extension modules linked in some cases to the VHHs might obliterate their small size benefit and negatively impact this diffusion and internalization. Some VHHs have been shown to cross cell membranes and reach cytosolic targets. In this case, their internalization has been linked to their basic isoelectric point and positive surface charges [30,31,32,33]. Even though the precise mechanism is still unknown, an adsorptive-mediated endocytosis could be hypothesized. Brain penetration is going one step further than cell penetration as the molecule must enter the endothelial cells forming the blood–brain barrier and then cross the membrane of the abluminal endothelial cell surface to reach the brain parenchyma. This is the subject of this review.

### 2.2. VNAR

In 1995, heavy-chain antibodies (HCAb) were also discovered in sharks [34]. These antibodies are composed of two heavy chains made of five constant domains (CNAR1, CNAR2, CNAR3, CNAR4 and CNAR5) (Figure 1). VNARs are the variable domain of these antibodies. As for VHHs, VNARs bear full antigen recognition properties. The main difference of their variable domain is the absence of complementary-determining region 2 (CDR2) leading to only two CDRs. Even though VNAR are lacking one CDR, they can still recognize numerous antigens thanks to a higher variability in CDR1 and a longer CDR3 [35]. These characteristics make them the smallest antibodies with a molecular weight of only 12 kDa.

VHHs and VNARs small sizes allow them to be good candidates as therapeutics, diagnostics and transporters. Moreover, this small size allows them also to reach buried epitopes, facilitating the discovery of mouse–human cross-species reactive sDAbs, a feature not always accessible with conventional IgGs [36].

## 3. Single-Domain Antibodies Crossing the Blood–Brain Barrier (BBB)

### 3.1. Mechanisms

Several mechanisms have been reported for VHHs that cross the BBB. These ways are either direct by using endogenous brain transport mechanisms or indirect relying on BBB opening either in disease-free or disease-dependent state. 

#### 3.1.1. Receptor-Mediated Transcytosis 

The first mechanism used by some VHHs to go through the BBB is the same reported for some conventional antibodies: receptor-mediated transcytosis (RMT). Receptors mostly used to deliver conventional antibodies into the brain are transferrin and insulin receptors [37,38,39] but others such as lipoprotein-related proteins [40,41] or IgF1 receptors have also been reported [42,43]. Fusions of insulin and transferrin receptors antibodies to iduronidase lysosomal enzyme are currently evaluated in the clinics against mucopolysaccharidosis [44,45]. The first VHHs found to perform receptor-mediated transcytosis across the BBB are FC5 and FC44. These two VHHs discovered in 2001 were obtained from a naive llama phage-displayed library followed by a panning on human endothelial cells forming BBB [46]. The aim was to identify VHHs recognizing these cells and able to transmigrate across them. Their brain uptakes have been quantified and shown to be more than 10 times higher than controls [47]. The VHH FC5 internalization mechanism has been determined as a clathrin-dependent receptor-mediated transcytosis [48]. The protein involved is the transmembrane domain protein 30 A of α(2-3)-sialoglycoprotein and the use of this domain to obtain molecules for brain delivery has been patented [49].

More recently, Stanimirovic et al. patented VHHs recognizing the insulin-like growth factor 1 receptor and transmigrating across the BBB by RMT [50]. The VHH name is insulin-like growth factor 1 receptor 5 (IGF1R5) and its transcytosis has been tested in vitro on a BBB model and in vivo on mice. The VHH recognizes IGF1R at the luminal side of endothelial cells to transcytose across the cells and be released in the brain environment on the abluminal side of the BBB. Its humanized and Fc-fusion versions have shown an improved passage through the BBB. We show in the second part of this review that this VHH can be used to deliver a cargo into the brain. 

VHHs FC5 and IGF1R5 have been used as positive controls to validate the new human stem cell BBB model described by Ribecco-Lutkiewicz et al. [51].

Several VNARs directed against transferrin receptor (TfR1) have also been designed by Ossianix, to cross the BBB by receptor-mediated transcytosis. The two main VNARs developed against TfR1 are B2 and TXB2. To increase their half-lives, they have been linked to Fc domains. B2-Fc [52] and TXB2-Fc demonstrated up to 10–20-fold brain enhancement versus controls in PK studies. These sDAbs have mostly been used as transporters of biotherapeutics into the brain demonstrating pharmacological effects that are reviewed in the second part of this paper.

#### 3.1.2. Adsorptive-Mediated Endocytosis

Two decades ago, positive charges addition at the protein surface was already reported as enabling brain penetration [53]. This has been applied to IgGs after cationization by covalent coupling with hexamethylenediamine or putrescine raising their isoelectric point largely above 7 favoring membrane crossing through the process of adsorptive transcytosis [54]. One peculiar aspect of VHHs is probably that, owing to their high content in exposed hydrophilic amino acids, they often appear to be produced with largely basic isoelectric points. This is the case of VHH E9 [31] recognizing glial fibrillary acidic protein (GFAP), an astrocyte marker, shown to cross the BBB after in vivo administration of a fluorescent derivative. These results have been obtained on mice after intracarotid perfusion or lateral tail vein injection with either VHH alone or VHH-eGFP. The VHH has been detected on astrocytes, meaning that, after crossing the BBB, the VHH reaches cell cytosol and its target. The same team has published another study in which two other VHHs cross the BBB [32]. Both VHHs have a basic isoelectric point. The first VHH is directed against the peptide Aβ42 in amyloid plaques (VHH R3VQ) and the other one against the phospho-tau (VHH TauA2). After intravenous injection with 10–20 mg/kg of VHH in PS2APP mice, in vivo two-photon microscopy revealed that amyloid plaques and neurofibrillary tangles were stained. These two VHHs could be used in imaging Alzheimer’s disease main lesions in a transgenic amyloid peptide precursor (APP) mouse model. A complementary unpublished pharmacokinetics study has indicated that 2 h after intravenous injection, 0.5% of the injected dose VHH TauA2 was found in the brain. Even though this percentage is higher than for classical antibodies, which are found in the brain at 0.01–0.4% [55], this penetration rate remains low. In addition, since VHHs half-lives are much shorter than the ones of conventional antibodies, higher or more frequent dosing will be necessary to compensate these low rates. The precise mechanism has not been determined but the importance of positive charges and basic isoelectric point has been noticed such as for other VHH’s internalization [30,31]. Another anti-Aβ amyloid VHH carrying 18 positive charges has been shown to cross the BBB using an active transport in an in vitro model [56].

#### 3.1.3. Carrier-Mediated Transcytosis

Nanotechnologies have largely been reported to be able to cargo small molecules or oligonucleotides into the brain if they are targeted to the brain by specific ligands [57] but examples with antibodies are rare. The concept has been specifically applied to an anti-Aβ amyloid VHH after encapsulation in liposomes decorated with glutathione (GSH) [58]. This tripeptide was believed to use a specific carrier that was recently reported to be the N-Methyl-D-Aspartate receptor (NMDAR) [59]. This VHH pa2H encapsulated into liposomes was able to stain amyloid plaques into the brain after intravenous administration. The brain exposure of the encapsulated VHH was improved compared to the VHH alone from 0.001% of injected dose (ID) to 0.015% and 0.094% of ID in wild-type (WT) and APP/PS1 transgenic mice, respectively [60]. GSH-liposome encapsulation also allowed decreasing the clearance of the VHH from the blood.

#### 3.1.4. BBB Opening 

The two previous paragraphs report on sDAbs crossing the BBB by making use of endogenous mechanisms by which the brain imports its proteins. This formidable challenge represented by BBB can also be circumvented by other means. Low energy ultrasounds in parallel to injected microbubbles have the potential to temporarily open the BBB [61]. Even though a few reports have applied the technology to antibodies [62,63], we are not aware of application to sDAbs. Temporary BBB disruption can also be induced by a few agents such as mannitol provoking an osmotic opening of the BBB which can lead to increased brain exposure. This was already applied to Fabs and IgGs, showing that osmotic BBB disruption significantly increased monoclonal antibody delivery to the brain [64]. However, the extent of enhancement is modest. Osmotic disruption has been applied to the anti-gelsolin VHH Nb11 [65] designed for the potential treatment of hereditary gelsolin amyloidosis, an autosomal dominantly inherited amyloid disorder. PET imaging showed that mannitol BBB opening allowed an increase of the VHH in the brain of around 2.5-fold, however the VHH was injected intraarterially [66]. This mode of administration has already demonstrated an advantage over intravenous administration regarding brain exposure [67].

#### 3.1.5. BBB Integrity Modified by Diseases

In some diseases such as in multiple sclerosis [68] or in cerebral malaria that is deadly [69], the BBB can be altered. As with mannitol, this alteration might allow molecules to reach the brain. In African trypanosomiasis late stage, also called sleeping sickness, *Trypanosoma brucei* parasite alters the BBB and induces brain inflammation [70]. The VHH Nb-An33 recognizing a surface glycoprotein of this parasite has been developed and tested on rat. The nanobody has been injected at 4 mg/kg in normal rat versus rat with encephalitis comparable to late stage sleeping sickness. The VHH showed capacity to reach the brain in both cases. However, in a rat brain from a late stage of encephalitis, VHH found in hippocampal extracellular fluid was doubled compared to a normal rat brain. The VHH concentration measured in the brain was 50 ± 21 ng/mL in control rat and 131 ± 63 ng/mL in the late stage disease model [71], showing slightly better exposure linked to altered BBB.

#### 3.1.6. Intranasal Delivery: Another Brain Delivery Option for VHH

Intranasal delivery of therapeutics involves spraying therapeutics into the upper part of the nasal cavity to enable them to follow the olfactory axon bundles directly into the brain [72,73]. This non-invasive, needle-free and painless method has already allowed brain exposure of a radiolabeled IgG in rats [74]. It has also been used for VHH delivery into the brain. Ablynx has patented intranasal delivery of therapeutic polypeptides and proteins including VHHs [75]. This strategy has been applied to a VHH against transthyretin protein, used as a research tool, allowing a positive brain distribution into several areas [76].

### 3.2. Single-Domain Antibodies as Therapeutics or Diagnostics in Central Nervous System (CNS) Diseases

Even though we show in the previous section that several sDAbs have demonstrated enhanced brain exposure, no VHH or VNAR are presently in clinical development in a CNS application except for one VHH currently in phase II development [77] in breast cancer brain metastasis. A preclinical study had shown an increase of survival rate with this VHH CAM-H2 [78]. A phase I clinical study has confirmed the safety of this radiolabeled VHH in patient, allowing its use as a therapeutic or as a diagnostic tool [79]. An ongoing clinical trial aims at showing the use of this anti-HER2 VHH as a diagnostic tool for human epidermal growth factor receptor 2 (HER2)-positive brain tumors [80]. This might be enabled by local impairment of the BBB that can occur in brain cancers and in diseases such as stroke, Parkinson’s disease, AD and multiple sclerosis even if the extent and duration of the leakage is highly variable [81]. This can be the case also in some high-grade gliomas, where some degree of parenchymal penetration of large molecules (such as antibodies) is likely in areas in which contrast extravasation is detected by imaging, evidencing disruption of the BBB. However, this disruption is heterogeneous and might mostly represent microscopic disease foci behind an intact BBB [82].

Several examples describing sDAbs used as either therapeutics, diagnostics, theranostics or transporters are found in the literature and will be the object of the rest of this paper. We include in this review only the examples where the sDabs designed for a CNS indication or imaging purpose have also generated positive data after in vivo evaluation illustrating thereby that they have reached the brain.

#### 3.2.1. VHHs as Brain Therapeutic Tools

As rabies virus can target the brain leading to death-causing inflammation, a therapeutic molecule able to reach the brain and neutralize the virus would be key. Conventional immunoglobulins are currently used but an additional brain neutralizing effect would be desirable. VHHs recognizing the glycoprotein of rabies virus have been developed and shown to be able to neutralize the virus in vitro [83]. After extending their half-lives by coupling with an albumin binding VHH, they were tested in vivo in mice. After preliminary validation of the effect upon direct brain injection, the VHHs were tested by intraperitoneal injection. Compared to a conventional antibody that allowed a prolonged survival of two days without rescue, these VHHs constructs allowed more than nine days of prolonged survival with 43% of rescue. However, their brain exposure corresponded to 0.1% of their plasma exposure, the same rate as for conventional antibodies. The authors concluded that this difference could be due to the neutralizing power of the VHHs rather than their ability to cross the BBB [84]. Even if the brain penetration could be optimized, this VHH could be a potential therapy for rabies disease. 

The VHH PrioV3, designed to target brain misfolded prion protein causing prion disease, has been reported to cross the BBB in vitro and in vivo. After crossing the BBB, this VHH was able to prevent prion replication by reaching neuron cytosol [85]. The mechanism of BBB crossing involves prion protein at the surface cell membrane and clathrins [86].

#### 3.2.2. VHHs as Brain Diagnostic Tools

In addition to a therapeutic use, VHHs reaching the brain can also be used as diagnostic tools. Rutgers et al. [87] developed, from immune and non-immune libraries, VHHs recognizing either parenchymal or vascular beta amyloid. They were evaluated to image brain lesions of Alzheimer’s disease. Among the VHHs obtained, ni3A was actively passing through an in vitro model of BBB [56]. The BBB passage was further evaluated in vivo for VHH ni3A and pa2H obtained from the first screening. The VHHs were radiolabeled with two different labels (^111^In and ^99m^Tc) and their brain uptakes monitored. ^99m^Tc-labeled VHH ni3A and pa2H crossed the BBB and were quantified in the cerebellum at 0.053% ID/g and 0.04% ID/g, respectively. Even if ^99m^Tc-VHH allowed a quantification into the brain, the rate was too low for imaging [88]. ^111^In-pa2H quantification did not differ from the control with ~0.001% ID/g. A better knowledge of transport mechanism of these VHH into the brain would allow an optimization of BBB crossing to use as a diagnostic tool. The VHH pa2H has been engineered with a Fc domain to increase half-life and decrease blood clearance. This new construction allowed an increase half-life but did not improve brain uptake keeping the quantified VHH at 0.001% ID/g [89]. However, the ^111^In radiolabel that showed less brain uptake than ^99m^Tc in the previous in vivo study was used. To improve brain exposure, the VHH was encapsulated into GSH-decorated liposomes which led to increased exposure up to 0.025% ID [60]. Even though the rate of BBB passage is not high, ni3A and pa2H could be used to image amyloid plaque as it has been shown on APP/PS1 amyloid transgenic mice [88].

VHH E9 and R3VQ, described previously as brain penetrant could also be used as a diagnosis marker of Alzheimer’s disease as they recognize its main lesions [32]. VHH R3VQ recognizing amyloid plaques has been coupled to gadolinium, a magnetic resonance imaging (MRI) contrast agent. Immunohistochemistry showed positive staining of amyloid plaques validating the BBB passage [90]. Even if MRI visualization of amyloid plaques was obtained in vitro, no labeling of plaques was obtained in vivo on live mice, suggesting that the amount of VHH in the brain was too low. However, these results are promising for a potential use of R3VQ for amyloid plaques detection on patients.

A nanobody directed against vascular endothelium growth factor (VEGF) has shown a BBB passage optimized by mannitol treatment and intra-arterial administration instead of intravenous injection. The improved brain uptake allowed imaging of the brain by positron electron tomography (PET) using ^89^Zr as radioactive tracer [66].

The brain can also be imaged in the case of brain cancer. Iqbal et al. [91] used a VHH against EGFR to image glioblastoma in the brain. Brain images are better when VHH named EG2 was coupled to an Fc domain.

#### 3.2.3. VHHs as Brain Theranostic Tools

We have previously seen that VHHs can be used as therapeutic and diagnostic tools. We now discuss the potential of VHHs as theranostic tools, a combination of both diagnostic and therapeutic purposes. Puttermans et al. [92] described this option with VHH 2Rs15d targeting human epidermal growth factor receptor 2 (HER2). The overexpression of this protein is associated with several cancers such as breast cancer [91]. The VHH has been coupled with three different radionuclides: ^111^In, ^225^Ac and ^131^I. Each radionuclide coupled to VHH was tested one by one or in combination and the impact on survival was evaluated and compared to trastuzumab, a conventional antibody targeting HER2 and already approved for HER2-positive cancer treatment. This VHH could reach the brain, recognize and detect HER2-positive brain lesions caused by breast cancer metastases and improve the survival rate by killing tumor cells.

#### 3.2.4. VHHs and VNARs as Transporters for Therapeutic Molecules

VHHs or VNARs passing through the BBB by RMT could successfully be used as transporters. FC5 was first used for brain delivery of an anti-cancer drug doxorubicin, that cannot reach the brain on its own because of efflux pumps [93]. The VHH was either used as a monomer (FC5) or as a pentamer (named P5) [94]. VHH FC5 was used to decorate liposomes encapsulating doxorubicin, as shown in Figure 2. After intravenous injection of 6 mg/kg, doxorubicin was quantified in brain parenchyma. The functionalization of the liposome modestly increased brain uptake of doxorubicin originally from 100 ng/g of tissue to 200 and 350 ng/g with FC5 and P5, respectively [95].

Farrington et al. [96,97] also showed that FC5 can deliver a cargo such as dalargin into the brain by using different constructions. They have reported that FC5 alone or coupled to an Fc domain can deliver peptides leading to more than 30-fold improvement compared to the control without VHH. In these constructions, the Fc domain allowed an increase of half-life leading to an improvement of VHH brain delivery, thus of the peptide. 

VHH FC5 could also be dimerized with a conventional antibody such as an anti mGlutR1, a potential target receptor found in the CNS. This led to an increase of brain delivery in vivo of about 20-fold, validating again the possibility for FC5 to deliver a cargo into the brain [98].

Stanimirovic et al. [50] also illustrated the potential of the VHH IGF1R5 as a transporter of galanin. This neuroactive peptide can induce analgesia after binding to its receptor in the brain. However, it is inactive when given by peripheral route as it does not penetrate the BBB on its own. The analgesic effect was obtained when galanin was coupled to IGF1R5 and injected into a hyperalgesia inflammatory rat model.

A VHH against vascular cell adhesion molecule 1 (VCAM1) named VHH against VCAM1 (VCAMelid) has recently shown its ability to deliver SOD-1 enzyme into the brain. This delivery was either by a direct coupling to VHH or by enzyme encapsulation on VHH-decorated liposomes. The liposomes delivery system seems to be analogous to the FC5 liposomes presented previously (Figure 2). VCAMelid was reported to reach the brain on its own and allowed brain delivery of SOD1 both in naive and inflammatory mice model. Brain exposure of VCAMelid was higher in the disease model of mice with local injury compared to naive mice with brain uptake alone at around 0.8% ID/g and 0.2% ID/g, respectively. There was also a difference between the conjugated construct and the liposomal formulation with SOD1 brain uptake in injured mice, of around 1.2% ID/g and 2% ID/g, respectively [99]. These brain uptake constructions are the most efficient found and described in this review with a great difference compared to the control which is around 0.1% ID/g.

Recently, a new liposome construction was designed for brain delivery of anti-cancer drugs targeting brain metastases. The liposomes were decorated with TfR-binding peptides and anti-program death-ligand 1 (PDL1) VHHs. It has been shown that brain delivery of the liposome-encapsulated simvastatin/gefitinib drug combination was driven by both targets (TfR and PDL1) present on endothelial cells of BBB bearing metastases. With both anti-TfR peptide and anti-PDL1 VHHs these liposomes allowed brain delivery reducing brain tumor and improving survival [100]. The liposomes solely targeted with anti-PDL1 nanobody mainly showed macrophages targeting allowing cell internalization of the liposome but were not efficient enough without the anti-TfR peptide to reach the brain [101].

Prehaud et al. patented a basic VHH with no identified brain target for which in vitro experiments suggest a potential to cargo several different effectors such as neuroprotective peptides into the brain justifying to protect their use for brain transport [102,103]. 

Recently, Vect-Horus, a biotechnology company based in Marseille in France working on designing vectors to go through barriers, has patented VHHs against transferrin receptors. These VHHs display comparable affinity to human and rodent TfR and could be useful for shuttling therapeutics or imaging agents into the CNS [104].

Ossianix is reporting preclinical research in several fields of CNS such as pain, multiple sclerosis, CNS lymphoma or glioblastoma with fusions of their VNARs with several therapeutic proteins. The VNAR named B2 described by Wicher et al. [52] has shown its ability to deliver rituximab, an anti-CD20 antibody for peripheral B cell lymphoma treatment, into the brain [105]. Brain exposure was more than ten times higher than the naked antibody. The TXB2 VNAR was also fused to the neurotensin peptide demonstrating a rapid, sustained CNS exposure and robust pharmacological activity after injection of 1.875 mg/kg in mice [36]. On the other hand, while TBX2-VNAR-Fc fusion demonstrated 20-fold brain exposure improvement vs. control VNAR-Fc at 18 h post-injection in wild-type (WT) mice, when the TXB2-VNAR was fused to the N-terminal light chains of the anti-amyloid antibody bapinezumab, the brain exposure enhancement was three-fold at the same time and up to six days after injection [106]. PET imaging and autoradiography evidenced more parenchymal Bapi-TBX2 compared to Bapi. It has to be noted that, in this case, because of Fc-fusion domain, the benefit of the VHHs small size is lost and their brain penetration mostly relies on the transferrin receptor-mediated mechanism.

## 4. Conclusions

In this review, we describe VHHs and VNARs that can reach the brain directly (RMT and AMT) or indirectly (brain targeted liposomes, BBB-opening and disease). We try to provide the best overview of already used VHHs and VNARs into the brain. Although several others have been developed against brain targets, their brain exposure has not been demonstrated or quantified. All these delivery systems using either VHHs or VNARs can be used for brain targeting of active principles that could lead to promising new treatments for diseases with no current treatment.

## Figures and Tables

**Figure 1 pharmaceutics-12-00937-f001:**
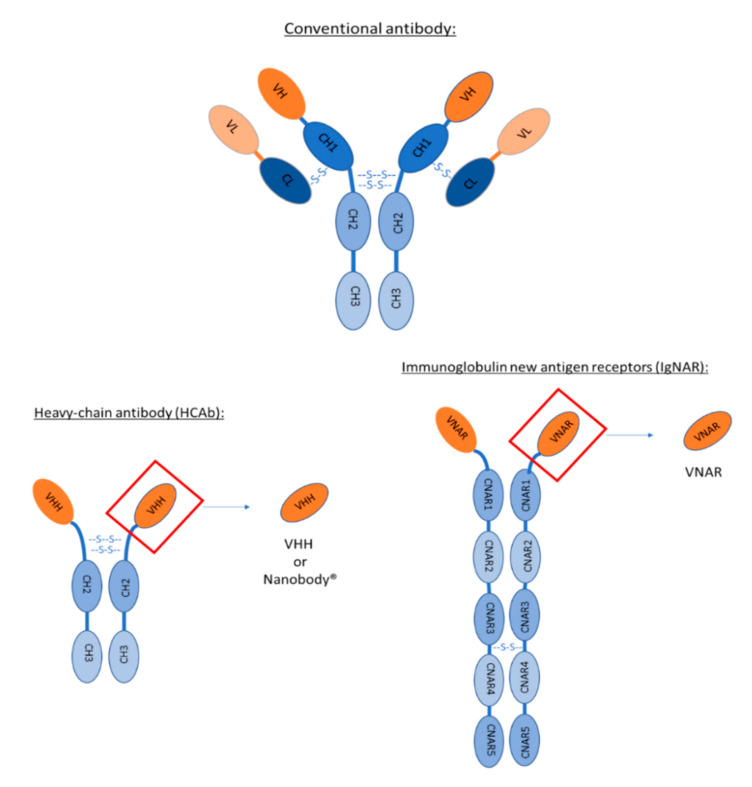
Schematic representation of conventional antibody versus heavy-chain antibody found in Camelidae versus immunoglobulin new antigen receptor found in sharks. VHH is the variable domain of heavy-chain IgG2 and IgG3. The difference between these two IgG is the length of the hinge (region between VHH and CH2). VNAR is the variable domain of IgNAR.

**Figure 2 pharmaceutics-12-00937-f002:**
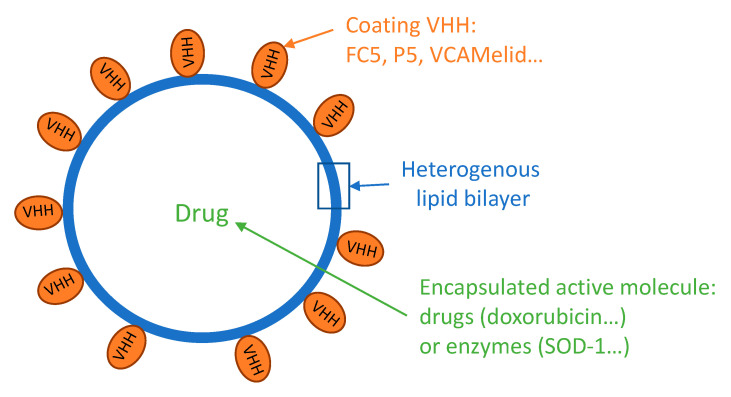
Encapsulated active molecule (drug or enzyme) into VHH-coated liposomes for brain delivery.

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
