# Peer review of "Brain Delivery of Single-Domain Antibodies: A Focus on VHH and VNAR"

_pharmaceutics, 2020, doi:10.3390/pharmaceutics12100937_

Round 1

Reviewer 1 Report

This manuscript by Pothin et al is a review of how single domain antibodies such as VHHs and VNARs can be used for brain delivery. The review is thorough and provides a great overview of the uses for single domain antibodies as therapeutics themselves, or as tools to deliver other therapeutics. They also discuss mechanisms by which these antibodies reach the brain, by both direct (e.g. receptor-mediated transcytosis) or indirect (e.g. targeted liposomes) methods. I recommend publication after correcting the following minor error.

On line 100, the text labels the five constant domains of VNAR antibodies as CH1 through CH5 and references Figure 1. However, in Figure 1 one, VNAR constant domains are named CNAR1-5. This should be changed, one way or the other, to be consistent.

Author Response

Dear Madam / Dear Sir,

Thank you for your review. We have taken into account your comment and we have replaced in the text all constant domain by “CNAR” instead of “CH” for antibodies found in sharks.

Regards,

Elodie Pothin

Reviewer 2 Report

Brain delivery of single-domain antibodies: a focus on VHH and VNAR

Elodie Pothin Dominique Lesuisse and Pierre Lafaye

The current review is one that is focussed specifically on single domain antibodies and their development as modalities that can cross the blood brain barrier (BBB).

This is a useful review that focusses on the use of single domain antibodies that have been proven to cross the BBB for therapeutic and diagnostic purposes.

General points

Given the purpose of this review is to highlight single domain antibodies that are being developed for CNS (specifically in the brain) and that the authors actually conclude with “However, several others have been developed against brain targets, but their brain exposure have not been demonstrated or quantified” I would strongly advise removing any reference to any sDAB that has not been proven to cross the BBB in a peer-reviewed journal. Some of the company content in Table 1 has not been proven and should be deleted as it lacks robust scientific evidence.

Another key factor that should be elaborated further within this review are the benefits of the sDAB used in each example. The reason for this comment is that the authors state multiple times the benefit of the small size of these domains in penetrating and crossing tissue/membranes (ie BBB) however in the case of Ossianix for example they have not developed any VNAR that is not fused to an Fc or a whole antibody so there are not examples of the benefit of size whereas some of the VHH are truly small even when fused to an HSA binding VHH. This should be made clear to the reader.

A general observation is that a number of the references are reviews themselves. This hinders the reader finding the actual data to support the claim. It is favourable to avoid references to reviews as this just becomes a multi-layered review of reviews and to reference the original manuscript.

There are also quite a few grammatically weak sentences – I do not intend to highlight all of these as the editors should go through this in depth. I will point to a few below but do not feel this is responsibility of the reviewer.

Specific points

Line 26; “as diagnosis tools”. The authors here are presumably referring to in vivo real time imaging or similar. Given a diagnostic tool can be used in vitro and ex vivo and half life is not an issue, it would useful for them to define which aspect or type of diagnosis to which they are referring so the reader can appreciate it that the long life does have an impact. Given this is a review – a reference to this issue in the literature would be appropriate.

Line 33; “preventing most molecules to penetrate” – grammatically poor

Line 70 – registered as opposed to “deposited”

Line 83 – “affinity is similar than antibodies”. Suggest change “than” to “to”.

Line 98 – poor grammar

Line 102 – “capacity to recognize a maximum of antigens”. Unclear what this sentence means.

Line 145 – poor grammar

Line 197 – poor grammar

Line 194 – given all the subtitles in this section refer to a type or mechanism for delivery across the BBB, it would be better to change the subtitle “Disease”, which is very vague, to fit with this format. Presumably something referencing that certain diseases result in the BBB opening or reducing the integrity of the BBB.

Table 1 – given the review states the authors are focussing on CNS single domain antibodies that can reach the brain ie there is published scientific proof, all references to company pipeline products that have no peer-reviewed data to prove their ability to cross the BBB and reach the brain should be removed. Otherwise the impact of the review is greatly reduced.

Line 348 – Statement about Prehaud et al with no reference. Please include a reference in the paragraph.

Author Response

Dear Madam / Dear Sir,

Thank you for your review. We have considered all of your comments. You will find attached the modifications we have added in the revised version of the manuscript and the answers to your comments.

Best regards,

Elodie Pothin

Reviewer 3 Report

authors present an interesting review of an important topic

few comments listed below

  • small MW of compounds described in the review leads to truncated half life which is a significant concern as it will require more frequent dosing. As a solution authors cite use of life extension "module". This will, of course, increase the size of the molecule which includes more than just the MW. It will also impact ability of the molecule to penetrate tissue barriers, including BBB. The smaller size nanobodies (e.g.) can indeed penetrate tissue better, as authors appropriately state. But with the extension module this advantage may be lost in part or all together. Authors need to comment on this disadvantage and discuss balance of half life extension vs. tissue penetration challenge 
  • authors discuss various modifications to VHH or other compounds and propose that these changes may improve on the tissue penetration properties. Authors should also comment on the immunogenicity risk that can be easily introduced by either using domain of Ig or non-human sequence and by adding modifications to the protein sequence, even if fully human. All three are known to cause a robust immune response in the clinic which can be detrimental to the drug development 
  • the penetration values cited for VHH protein (0.5%) even though higher than the same referenced for full length Ig, are still very low and will require a significant dosing regiment particularly considering short half life of the molecule. Authors need to comment      

Author Response

Dear Madam / Dear Sir,

Thank you for your review. We have taken into account all of your comments. Here are the modifications we have added in the revised version of the manuscript :

  • A sentence has been added line 120 to deal diffusion problem when life-extension added:
    “Their small size allows them to diffuse more than four times better in tissues [27,28] and tumors [13] than antibodies. However, half-life-extension modules linked in some cases to the VHHs might obliterate their small size benefit and negatively impact this diffusion and internalization. Some VHHs have been shown to cross cell membranes and reach cytosolic targets.”
  • A sentence has been added line 106 after description of caplacizumab:
    “This VHH has been developed against von Willebrand factor and is available for patients with acquired thrombotic thrombocytopenic purpura [15,16]. VHHs low immunogenicity has been evaluated with native VHHs. In this review, we will also deal with mutated or complexified VHHs that might have different immune responses that will have to be checked before any clinical use. Compared to conventional antibodies, VHHs are more stable and can refold after denaturation [17]”.
  • A sentence has been added line 234 to cite that 0,5% is a low penetration rate:
    “A complementary unpublished pharmacokinetics study has indicated that two hours after intravenous injection, 0.5% of the injected dose VHH TauA2 was found in the brain. Even though this percentage is higher than for classical antibodies which are found in the brain at 0.01% to 0.4% [53], this penetration rate remains low. In addition, since VHHs half-lives are much shorter than the ones of conventional antibodies, higher or more frequent dosing will be necessary to compensate these low rates. The precise mechanism has not been determined but the importance of positive charges and basic isoelectric point has been noticed such as for other VHH’s internalization [29,30].”

Regards,

Elodie Pothin

Round 2

Reviewer 2 Report

I am happy with the amendments made to this manuscript and am pleased that the authors have composed this review based on peer-reviewed comparable data and information which makes a valuable addition to the field. 

Reviewer 3 Report

no additional comments